



# Data reporting and sharing for ocean alkalinity enhancement research

Li-Qing Jiang[1,2], Adam Subhas[3], Daniela Basso[4], Katja Fennel[5], and Jean-Pierre Gattuso[6,7]

[1]Cooperative Institute for Satellite Earth System Studies, Earth System Science Interdisciplinary Center, University of Maryland, College Park, Maryland 20740, United States.

[2]NOAA/NESDIS National Centers for Environmental Information, Silver Spring, Maryland 20910, United States.

[3]Department of Marine Chemistry and Geochemistry, Woods Hole Oceanographic Institution, Woods Hole, Massachusetts
02543, United States

[4]Department of Earth and Environmental Sciences, University of Milano-Bicocca, Piazza della Scienza, 4, 20126 Milano, Italy.

[5]Department of Oceanography, Dalhousie University, Halifax, Nova Scotia, B3H 4R2, Canada.

[6]Sorbonne Université, CNRS, Laboratoire d'Océanographie de Villefranche, F-06230 Villefranche-sur-Mer, France.

[7]Institute for Sustainable Development and International Relations, Sciences Po, F-75007 Paris, France.

*Correspondence to*: Li-Qing Jiang (Liqing.Jiang@noaa.gov)

**Abstract.** Effective management of data is essential for successful ocean alkalinity enhancement (OAE) research, as it guarantees the long-term preservation, interoperability, discoverability, and accessibility of data. OAE research generates
various types of data, such as discrete bottle measurements, autonomous measurements from surface underway and uncrewed platforms (e.g., moorings, Saildrones, gliders, Argo floats), physiological response studies (e.g., laboratory, mesocosms, and field experiments, and natural analogues), and model outputs. This chapter covers data and metadata standards for all these types of OAE data. As part of this study,  existing data standards have been updated to accommodate OAE research needs, and a completely new physiological response data standard has been introduced. Additionally, an
existing ocean acidification metadata template has been upgraded to be applicable to OAE research. This chapter also presents controlled vocabularies for OAE research, including types of studies, alkalinization methods, platforms, and instruments. These guidelines will aid OAE researchers in preparing their metadata and data for submission to permanent archives. Finally, the chapter provides information about available data assembly centers (DACs) that OAE researchers can utilize for their data needs. The guidelines outlined in this chapter are applicable to ocean acidification research as well.



# 1 Introduction

Data management plays a crucial role in bridging the gap between field observations and subsequent research based on these data (Brett et al., 2008). It is an essential component of ocean alkalinity enhancement (OAE) research to help evaluate its potential environmental risks and co-benefits, understand its effectiveness and scalability, and support its measurement reporting and verification (MRV) efforts for carbon credit accounting. Specifically, effective data management enables long-

term preservation of data, ensures compliance with uniform metadata and data standards, facilitates interoperability and compatibility, and enables data discovery and access (de La Beaujardière et al., 2010).

Long-term preservation can be achieved by publishing data in archives and preserving them in non-proprietary, archivable formats to ensure accessibility and retrievability for extended periods of time, spanning decades to even centuries. This helps prevent data loss or degradation caused by technological obsolescence, human errors, natural disasters, or other factors.

Datasets, unlike journal publications, are frequently revised or updated after they are released. This may occur as a result of additional quality control (QC), or the acquisition of additional data or metadata. While ensuring access to the latest version of a dataset is crucial, preserving previous versions is equally important. All historical versions should be retained on a permanent basis. Otherwise, research based on previous iterations of a dataset may become unverifiable.

Data standards are a set of rules and specifications that define how data should be stored, structured, and formatted (Berman

and Fox, 2013). Their purpose is to promote consistency and interoperability, reducing ambiguity in data exchange and interpretation. In oceanographic studies, data standards cover elements, such as the technical format for storing data, e.g., Microsoft Excel, Comma Separated Values (CSV), or NetCDF; standardized column header abbreviations and units; standardized methods for calculating certain variables; and missing value indicators. It's worth noting that the new XLSX format is based on OpenOffice XML, unlike the prior binary-based proprietary format of XLS. As a result, it's no longer a

proprietary format. By adhering to these standards, researchers can ensure that their data are organized, structured, and formatted in a way that allows for easy sharing, interpretation, and reuse.

Metadata refers to structured information that provides context and details about a dataset, such as its title, authors or creators, observed properties, instruments used, measurement and calibration details, uncertainty, and relevant keywords (Guenther and Radebaugh, 2004). It is often defined as data about data. Metadata serves two main purposes: first, it provides

users with detailed descriptions about a dataset, helping them understand it; and second, it offers search keywords that make the dataset findable and retrievable. Overall, metadata is a crucial aspect of data management and is essential for subsequent data use.

Controlled vocabularies are defined as lists of pre-defined and standardized terms (Zeng and Qin, 2008). The use of controlled vocabularies plays a very important role in effective data management, as it helps ensure that the data are

documented, findable, and accessible in a consistent way. By using a limited and standardized set of terms, controlled



vocabularies help improve metadata interpretation and data findability by eliminating spelling variations, synonyms, and other forms of variability. Additionally, controlled vocabularies help facilitate metadata interoperability between different systems, making it easier to exchange and integrate data between different organizations and platforms.

Data citation involves referencing a dataset for the purpose of attributing credit and facilitating access (TGDCSP 2013). It
not only enables data users to acknowledge and give credit to the producers of a dataset used in a research study or project, but also allows readers to access and use the dataset for additional research. Data citation plays an important role in promoting scientific reproducibility, and accountability, and facilitating data sharing and reuse. As data sharing becomes more prevalent, data citation is increasingly important for tracking the impact of data sets and ensuring that research is built on a strong foundation of credible and transparent data.

In essence, data management is a service aimed at fulfilling the data needs of the research community. Therefore, efforts to establish best practices, such as the creation of new data and metadata standards, and controlled vocabularies, should be driven by the needs and preferences of the research community. It is equally important for researchers to adhere to these guidelines when preparing high-quality metadata and data packages for submission to appropriate repositories. While this chapter also sheds light on recommendations and requirements for data assembly centers (DACs) to build customized data
management systems that meet the data needs of the OAE research community, the presented data and metadata standards primarily serve submission purposes. During the development of these data and metadata standards, we ensured that they have a wide range of applications in other research fields, including ocean acidification.

## 2 Data standards

Ocean Alkalinity Enhancement research encompasses a wide range of topics, resulting in various types of data. These
different types of OAE data can be classified into four categories: (a) discrete bottle-based measurements; (b) autonomous measurements from surface underway (e.g., surveys conducted on ships of opportunity, or SOOP), time-series (e.g., moorings), and uncrewed platforms (e.g., Saildrones, gliders, Argo floats); (c) physiological response studies, including laboratory, mesocosms, field experiments, and natural analogues; and (d) model outputs (Table 1). To ensure consistency and interoperability, it is recommended to use uniform data standards for each type of OAE data. For Category (b), two data
standards are available, one for surface underway measurements and the other for autonomous sensor data from uncrewed platforms, including moorings, Saildrones, gliders, Argo floats, etc. This is because the measurement of one of the key variables, the carbon dioxide fugacity ($f$CO$_2$), involves the use of two different systems, depending on whether it is monitored during underway operations or time-series mooring. Category (c) may also include abiotic responses, such as (but not limited to) saturation state thresholds for calcium carbonate precipitation, mineral dissolution rate studies, and CO$_2$
uptake efficiency determinations. In these cases, inorganic variables associated with these data standards should be sufficient to capture all of the relevant study details.



**Table 1.** Proposed data standards for the purpose of submitting common types of OAE data. A CTD rosette consists of a metal frame that houses a collection of sensors and water sampling bottles. The acronym CTD stands for Conductivity, Temperature, and Depth, which are the three primary variables measured by a CTD sensor. Furthermore, the rosette frame

can accommodate additional sensors to measure various oceanographic variables such as oxygen, chlorophyll a, and nitrate.

| Types | Definition | Data standards |
|---|---|---|
| Profile | The collection of discrete water samples from the ocean at specific locations and depths, using sampling bottles (e.g., Niskin). The samples are then analyzed in a laboratory to determine various oceanographic parameters, such as dissolved inorganic carbon, total alkalinity, dissolved oxygen, and nutrient levels. It can also refer to continuous measurements using autonomous sensors mounted on a CTD rosette. | Column headers for profile data[1]<br><br>Data file example[2]<br><br>Jiang et al. (2022) |
| Surface underway | Continuous measurements of oceanographic variables at the ocean surface using sensors, often in flow-through systems onboard research vessels or ships of opportunity (SOOP), to obtain real-time information about the ocean's physical and chemical conditions, such as temperature, salinity, and $fCO_2$. | Column headers for underway data[3]<br><br>Data file example[4] |
| Uncrewed platforms | Continuous measurements of oceanographic variables using autonomous or remotely operated platforms. Examples including time-series mooring, uncrewed surface vehicles (USV, e.g., Saildrones), profiling floats (e.g., Argo), autonomous underwater vehicles (AUV, e.g., gliders), instead of traditional manned research vessels. | Column headers for autonomous sensor data[5]<br><br>Data file example[6] |
| Laboratory experiments (Ch. 4.3) | A scientific investigation in which researchers manipulate parameters of the carbonate system in an aquarium of a laboratory to simulate future OAE conditions and observe the responses of one or multiple selected marine organisms. | |
| Mesocosms (Ch. 4.4) | Mesocosm studies are conducted in large, controlled outdoor tanks or enclosures that simulate natural conditions in the ocean. Mesocosms allow for the examination of multiple interacting factors that can affect the response of a community of marine organisms to OAE, including physical processes, such as hydrodynamics, and complex biological interactions, such as predator-prey relationships. | Column headers for physiological response data[7]<br><br>Data file example[8] |
| Field experiments (Ch. 4.5) | Field experimental studies typically involve the manipulation of total alkalinity and carbon dioxide levels in seawater at natural coastal or offshore sites and then monitoring the response of the surrounding marine ecosystem. | |
| Natural analogues (Ch. 4.2) | Natural gradients in carbonate chemistry and other relevant parameters can be used to study the sensitivity of the ocean system to future OAE conditions. The response of marine species and the broader ecosystem can be assessed in terms of their long-term acclimation and adaptation to enhanced total alkalinity. | |
| Model outputs (Ch. 4.6) | The outputs of mathematical models that simulate Earth system processes can be used to simulate real-world scenarios, and assess the impacts of different policies, among other purposes. | Balaji et al. (2018) |

[1]https://www.ncei.noaa.gov/access/ocean-carbon-acidification-data-system/support/profile.html.
[2]https://www.ncei.noaa.gov/access/ocean-carbon-acidification-data-system/support/profile.xlsx.





[3]https://www.ncei.noaa.gov/access/ocean-carbon-acidification-data-system/support/underway.html.
    [4]https://www.ncei.noaa.gov/access/ocean-carbon-acidification-data-system/support/underway.xlsx.
    [5]https://www.ncei.noaa.gov/access/ocean-carbon-acidification-data-system/support/autonomous.html.
    [6]https://www.ncei.noaa.gov/access/ocean-carbon-acidification-data-system/support/autonomous.xlsx.
    [7]https://www.ncei.noaa.gov/access/ocean-carbon-acidification-data-system/support/physiological.html.
[8]https://www.ncei.noaa.gov/access/ocean-carbon-acidification-data-system/support/physiological.xlsx.

Table 1 presents a list of recommended data standards for each type of the OAE data as mentioned above. This table serves as a reference for researchers and data managers to ensure that their data meet the required standards for long-term preservation, interoperability, and reuse. For all data standards, users may remove irrelevant columns and add necessary ones, as with the other data standards. The data standard for discrete bottle based observations is described in detail by Jiang

et al. (2022). The data standards for surface underway and autonomous sensor measurements are an update to what the community has been using over the last several decades.

The data standard for physiological response OA studies is developed as part of this study, covering laboratory experiments, mesocosms, field experiments, and natural analogues. It emphasizes the experimental setup while allowing users to document their own response variables. For biological variables, it is important to state the taxonomy (a taxon or a

community), upon which the variable is studied. For example, if the growth rate of a certain species of salmon is studied. The "variable/parameter" is growth rate, and "biological subject" is that species of salmon. One could group/capture organismal data in three forms: taxonomic, functional, and phylogenetic. It is recommended to use the species reference databases from the Catalogue of Life (https://www.catalogueoflife.org/), Integrated Taxonomic Information System (or ITIS, https://www.itis.gov/index.html), World Register of Marine Species (or WoRMS, http://marinespecies.org/), or Paleobiology

Database (PBDB, https://paleobiodb.org/classic/). For life stages, consider using an existing controlled vocabulary like https://vocab.nerc.ac.uk/collection/S11/current/.

Model outputs often involve extensive data volumes, reaching gigabytes or even terabytes, making it necessary to address standards for the operational provision of model data (i.e., making data available for weeks to years), separately from long-term or permanent archiving. The operational provision of model output data typically relies on three integrated standards:

network Common Data Form (netCDF) files, the Climate and Forecast (CF) metadata conventions, and the Open source Project for a Network Data Access Protocol (OPeNDAP) libraries for remote data access. NetCDF is an open-source software that was developed and supported by UCAR's Unidata program (http://www.unidata.ucar.edu/) since 1989. NetCDF enables the creation and dissemination of self-contained data files with metadata, using formats that are independent of any specific machine or system. It has long been a standard for generation of model outputs and climatological data products in the ocean

and climate modelling communities. The CF metadata conventions provide guidelines for encoding datasets in netCDF, specifying the reporting of space and time coordinates, units, variable names, and other relevant information (Hassell et al., 2017). CF-compliant netCDF files are advantageous due to their self-describing nature, eliminating the need for additional information to interpret their contents. CF is a living and open standard that encourages community participation in proposing



enhancements and reporting issues (https://www.cfconventions.org/). OPeNDAP, which is based on the Data Access Protocol
(DAP), allows remote access to CF-compliant netCDF files stored on webservers through a set of libraries, making compliant
data sets highly interoperable and findable. Furthermore, it enables users to request subsets of data without the need to transfer
potentially very large files when only a subset is of interest. Together, the netCDF-CF-OPeNDAP standard provides a high
level of readability and interoperability for model outputs, gridded data products (e.g., satellite observations), and ocean
observations (e.g., Argo). The evolution of these standards and their community-wide acceptance are discussed in Hankin et
al. (2010).

---

**Box 1:** Metadata elements that should be included in netCDF files generated out of a biogeochemical ocean model.
Information specific to OAE studies is indicated in italic font. Refer to Chapter 4.6 for more context.

- Identifier of permanently archived model code (e.g., in a Git repository) with version number that was used to create
  the dataset;

- Permanent reference to an accessible form of documentation describing model equations with concise description of
  key assumptions, e.g., *How is alkalinity represented? Are active feedback between biological process and carbonate
  system properties represented in the model? Is dissolution and precipitation of calcium carbonate considered and, if
  yes, how? Are exchanges between sediment and overlying water represented and, if yes, how?*

- Input data (e.g., model parameters, initial and boundary conditions) either directly or, where the data volume would
  be prohibitive, as a comprehensive description of how input data was generated and from which sources to enable
  reproduction including:
  - Domain bathymetry
  - Initial conditions of all state variables
  - Atmospheric forcing including $pCO_2$, any depositional fluxes, etc.
  - Lateral boundary forcing if applicable
  - River and groundwater inputs if applicable
  - Comprehensive list of all model parameters including physical and biogeochemical parameters, timestep,
    start time, output frequency
  - *Alkalinity and related inputs (dose, duration, timing, location)*
  - *Assumptions about solubility of particulate alkalinity feedstocks if applicable*

---

The netCDF-CF-OPeNDAP standard enables provision of model outputs in accordance with the FAIR principles (Wilkinson
et al., 2016), provided a few conditions are met. NetCDF-CF-OPeNDAP datasets can be Findable because machine-readable
metadata enable automatic discovery, Accessible because of the standardized communications protocol that is open and
universally implementable, Interoperable because of the standardized, machine-readable metadata and data and the ability to
subset and aggregate datasets, and Reusable because rich metadata using standardized naming conventions can be provided.





The necessary conditions for a netCDF-CF-OPeNDAP dataset to qualify as FAIR are that (a) it is openly available and has a globally unique and persistent identifier (e.g., a digital object identifier, or DOI), (b) data and metadata are registered and indexed in a searchable resource, and (c) data are described with rich metadata that include accurate and relevant attributes and remain accessible even if the data are no longer available. Box 1 lists attributes that should be included in netCDF files
generated out of a biogeochemical ocean model, including several that are specific to OAE research, for the output to be considered a richly documented dataset. Output from an Earth System model would have slightly different requirements regarding the atmosphere (e.g., atmospheric forcing would not apply).

The discussion thus far has focussed on the operational provision of model outputs, i.e., comprehensive datasets that may be available for periods of weeks to years. However, because of their large data volume, they are not amenable to long-term or
permanent archiving. Nevertheless, long-term archiving of model-related information in some form that makes datasets reproducible is required but not yet done routinely. We suggest the following as a best practice:

1.  Metadata should be permanently archived even for operational datasets (as mentioned above, this is required for a dataset to qualify as FAIR).
2.  Essential subsets of operational datasets should be permanently archived, although it may not be immediately clear
160       what these subsets should encompass. At a minimum, data subsets that would are required to support conclusions in publications should be archived. Expectations will likely emerge through a community process.
3.  Model code should be permanently archived (e.g., Git versions with DOI) and sufficient metadata should be provided so that investigators can reproduce all model inputs (including initial and boundary conditions, model parameters). This information should allow one, in principle, to reproduction the large model output datasets that
165       cannot be permanently archived.

## 3 Metadata template

Section 2 highlights the importance of including some specific metadata information in netCDF files generated out of ocean model outputs. Apart from fulfilling documentation purposes, such information plays a vital role in facilitating data discovery when utilizing the netCDF-CF-OPeNDAP standard for operational provision of model output data. However, for
long-term archiving purposes, DACs commonly implement an independent and comprehensive metadata template. Ideally, these templates should be universally applicable to all data holdings, ensuring comprehensive documentation and accurate discoverability.

Jiang et al. (2015) described a metadata template that can be universally applied to all major types of ocean acidification (OA) data. Its development was driven by the need to document laboratory experiments to study the physiological responses
of OA, which was a relatively new type of research at the time. The template benefited from the rich metadata management



experiences of the Ocean Metadata Editor (OME) as used by the Carbon Dioxide Information Analysis Center (CDIAC, Oak Ridge, Tennessee, USA). This is especially true for some of the metadata elements associated with ocean carbon parameters, e.g., carbon dioxide fugacity ($f$CO$_2$). It features a "variable metadata section", which allows the documentation of all ancillary metadata information of an observed oceanographic variable, e.g., its variable abbreviation, full name, unit,

instruments, uncertainty, etc., to be organized around the variable, thus enabling the documentation of rich metadata information for all observed properties. In addition, new metadata elements, e.g., observation type, *in situ* observation/manipulation condition/response variable, measured or calculated, biological subject, species identification code, life stage, etc. were introduced. As the template was being developed, a bottom-up approach was adopted and the authors worked with numerous OA scientists from around the world to ensure the produced template conforms to the needs and

preferences of the research community.

In this Chapter, an updated metadata template (Version 2.0) is presented to accommodate the documentation of data coming out of marine carbon dioxide removal (mCDR) research (Table 2). It specifically allows users to indicate the type of the OAE study, and indicate whether its treatment type is for future ocean acidification conditions or for ocean alkalinity enhancement experiments. It also has a new element for the name of the model. For the "people" sections, the address field

is split up into road address, city, state/province, zip code, and country for better machine-readability. The original title and abstract are replaced with "dataset title" and "dataset description", respectively, to make them distinguishable from the title and abstract of a peer-reviewed publication. The names of some other metadata elements were also changed to make them more self-explanatory. A new metadata element called platform type, which is backed with controlled vocabularies, is added to allow data users to filter the datasets based on the type of the specific observing platform. For example, in the future, a

user would be able to search for only Saildrone USVs-based measurements. For the funding information section, two new elements about the start date and end date of the project are added. Most importantly, an element is added to enable multiple datasets generated out of a research expedition or experiment to be linked to each other. Terms that were either obsolete or rarely used, e.g., spatial reference system, purpose, section, etc., were discarded.

**Table 2.** Selected components of the new metadata template. ICES is short for the International Council for the Exploration

of the Sea (https://vocab.ices.dk/?ref=315). For the latest version of the metadata template, refer to:

https://www.ncei.noaa.gov/access/ocean-carbon-acidification-data-system/oa-metadata-template/.

| No. | Component | Description | Controlled vocabularies |
|---|---|---|---|
| 1 | Submitter | Information about the submitter, including full name, institution, address, email, phone number, a persistent digital identifier (e.g., ORCID), etc. | https://orcid.org/ |



| 2 | Investigators | Information about the investigators, including their full names, institutions, addresses, emails, phone numbers, a persistent digital identifier (e.g., ORCID), etc. This component can be repeated as many times as needed. | ORCID: https://orcid.org/. |
|---|---|---|---|
| 3 | Author list for citation | Author list in the format of Lastname1, Firstname1 Middlename1; Lastname2, Firstname2 Middlename2; … The information will be used to compose data citation. It is not unusual for this list to be different from the investigators list. | n.a. |
| 4 | Dataset identifiers | This section covers information such as EXPOCODE, Cruise_ID, and digital object identifier (DOI) | n.a. |
| 5 | Dataset title | A brief descriptive sentence that summarizes the content of a data set. | n.a. |
| 6 | Dataset description | The abstract of a dataset is a brief summary that provides an overview of the dataset's content, purpose, and scope. It is used to provide context and background information to users who are interested in using the dataset. | n.a. |
| 7 | Types of study | There are several types of study designs that can be used to collect and measure oceanographic variables, or examine the physiological responses of marine organisms to OAE. Examples: surface underway, profile, time-series, laboratory experiment, mesocosms, field experiments, natural analogues, etc. See Table 4 for a list of controlled vocabularies for this element. | Table 4 |
| 8 | Treatment type | This element is only applicable if the above element is one of the physiological response studies. This element is designed to indicate whether a physiological response dataset is out of an ocean acidification (OA) or ocean alkalinity enhancement (OAE) study. | OA or OAE |
| 9 | Model name | [For model output dataset only] Name of the regional or global model, e.g., GFDL-ESM4.1. | n.a. |
| 10 | Temporal coverage | The start date and end date of the measurement in the format of YYYY-MM-DD. | n.a. |
| 11 | Bounding box information | Information such as the westernmost longitude, easternmost longitude, northernmost latitude and southernmost latitude of the study area (decimal degrees, negative for Western Hemisphere longitude). For laboratory experiment based studies, this field should be used for the location of the organism collection, and for mesocosms, field experiments, and natural analogues studies, this field should be used to indicate the location of the experiment. | n.a. |

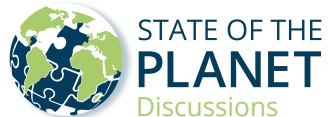

| 12 | Sea names | Names of the seas where the data collection takes place, e.g., Gulf of Mexico, Puget Sound, Baltic Sea, etc. | SeaDataNet C16 list (sea areas): https://vocab.seadatanet.org/v_bodc_vocab_v2/search.asp?lib=C16. |
|---|---|---|---|
| 13 | Location where biological subject was collected | Location where the organisms were collected, e.g., Puget Sound. | SeaDataNet C16 list (sea areas): https://vocab.seadatanet.org/v_bodc_vocab_v2/search.asp?lib=C16. |
| 14 | Location where the experiment was carried out | Descriptive words about where the experiment was carried out, e.g., Laboratory in Northwest Fisheries Science Center. | SeaDataNet C16 list (sea areas): https://vocab.seadatanet.org/v_bodc_vocab_v2/search.asp?lib=C16. |
| 15 | Platform type | Controlled vocabularies for the types of the platform (e.g., research vessel, ships of opportunity, fish vessel, oil tanker, mooring, Saildrone, glider, Argo float, etc). See Table 7 for a list of controlled vocabularies for this element. | Table 7 |
| 16 | Platform info | Detailed information about the specific platform, including name, ICES* platform code (if applicable), institution that owns this platform, and the country of the platform. | SeaDataNet Ship and Platform Codes: https://vocab.ices.dk/?ref=315 |
| 17 | Research project | Project, which the data collection is part of. For example, West Coast Ocean Acidification (WCOA) Project. | n.a. |
| 18 | Funding info | Information about the funders that supported the collection of this dataset, including the funding agency name, Project title, and Project identification, Project start date and project end date. | Research Organization Registry (ROR): https://ror.org/. |
| 19 | Supplementary information | Any additional information that cannot be accommodated in other metadata fields pertaining to this dataset. | n.a. |
| 20 | Publications describing the dataset | References of peer-reviewed publications that describe this dataset. It is recommended to use https://www.citationmachine.net/ to generate such references. | n.a. |
| 21 | Other datasets collected from this expedition | Sometimes, multiple datasets (e.g., one for chemical measurements, and another one for biological measurements) were produced out of the same research expedition. It is important to link them to each other. Please indicate the unique identification numbers of other published datasets that are related to this one in this field | n.a. |
| 22 | Variable metadata sections | See more details in Table 3. | n.a. |





Dataset title is a very important element of the metadata. It is often one of the few pieces of information a user can see in the search results. Thus, it is critical for data producers to create titles that are descriptive. It is recommended to follow the

template of "[observed properties] collected from [observation categories] using [instruments] from [research vessels or other platforms] in [sea names] during [research projects] from [start date] to [end date]. Here is one example: *"Dissolved inorganic carbon, total alkalinity, pH, temperature, salinity and other variables collected from profile and discrete sample observations using CTD, Niskin bottle, and other instruments from R/V Wecoma in the U.S. West Coast California Current System during the 2011 West Coast Ocean Acidification Cruise (WCOA2011) from 2011-08-12 to 2011-08-30".*

Dataset description is similar to the abstract of a publication, encompassing essential information on data collection and generation methods, the variables and attributes present in the dataset, as well as any limitations or restrictions on data usage. Moreover, it may provide instructions on accessing and utilizing the data. Here is an example of a well-crafted dataset description: *"This dataset contains discrete bottle (CTD profile) data of the first West Coast Ocean Acidification cruise (WCOA2011). The cruise took place aboard R/V Wecoma from August 12 to 30 in 2011. Ninety-five stations were occupied*

*from northern Washington to southern California along thirteen transect lines in the west coast of the United States. At all stations, CTD casts were conducted, and discrete water samples were collected with Niskin bottles. Inorganic ocean carbon variables, including DIC, TA, pH, as well as dissolved oxygen, and nutrients (Silicate, Phosphate, and Nitrate) were measured. The cruise was designed to obtain a synoptic snapshot of key carbon, physical, and biogeochemical parameters as they relate to ocean acidification (OA) in the coastal realm. During the cruise, some of the same transect lines were*

*occupied as during the 2007 West Coast Carbon cruise, as well as many CalCOFI stations. This effort was conducted in support of the coastal monitoring and research objectives of the NOAA Ocean Acidification Program (OAP)."*

One of the most important elements of the above metadata template is the "variable metadata section" (Jiang et al., 2015). It enables all ancillary information of a variable to be organized around the observed property (Table 3). Note that here "variables" refer to observed oceanographic properties, e.g., temperature, salinity, dissolved oxygen, pH, Nitrate, etc. They

should not be confused with other supporting variables such as EXPOCODE, Cruise_ID, year, month, day, yearday, longitude, latitude, depth, flags, etc. The latter elements are important for understanding the dataset, but the "variable metadata section" as described here is not applicable to them. Note that Table 3 shows the available metadata elements for a generic oceanographic variable. Customized variable metadata sections for ocean carbon variables (DIC, TA, $f$CO$_2$, and pH) allow additional information to be documented. Refer to the metadata template file for more details about these metadata

elements (https://www.ncei.noaa.gov/access/ocean-carbon-acidification-data-system/oa-metadata-template/).

Within the "Variable metadata section", the metadata element of "In-situ observation / manipulation condition / response variable" in Jiang et al. (2015) was replaced with "In-situ or manipulated". This change simplified this term, without





compromising the purpose of differentiating whether a term is an *in situ* observed variable, or a manipulated variable. New elements such as "Discrete or continuous", "Manipulation method", "Calculation method and parameters", "Sampling method", and "Analyzing method", "Calibration info", "QC steps taken", and "Weather or climate quality" were also added. Refer to Table 3 for their detailed descriptions. Metadata elements that were rarely used, such as "Purpose", "Sections (Cruise Legs)", "Duration (for experiment/settlement/colonization methods)", "Spatial reference system" were eliminated.

**Table 3.** Metadata elements available for each observed property in the generic variable metadata section. For the latest version of the metadata template, refer to: https://www.ncei.noaa.gov/access/ocean-carbon-acidification-data-system/oa-metadata-template/.

| No. | Element | Description | Controlled vocabularies |
|---|---|---|---|
| 1 | Variable abbreviation in the data file | The corresponding column header abbreviation of the variable in the data files, e.g., T, S, DIC, DO, etc. | Jiang et al. (2022): https://doi.org/10.3389/fmars.2021.705638, |
| 2 | Full variable name | Long name of the variable, e.g., water temperature, salinity, total dissolved inorganic carbon content, dissolved oxygen content, etc. | Table 1 in Jiang et al/ (2015): https://essd.copernicus.org/articles/7/117/2015/. |
| 3 | Variable unit | Units of the variable, e.g., degrees Celsius, μmol/kg, etc. | n.a. |
| 4 | Observation type | How the variable was observed, e.g., surface underway, profile, time series, model output, etc. For experimental data, this could be: laboratory experiments, pelagic mesocosms, benthic mesocosms, natural analogues, etc. See Table 4 for a list of controlled vocabularies for this element. | Table 4 |
| 5 | Discrete or continuous | Whether the reported results are based on discrete-bottled measurements or continuous sensor measurements. | Discrete vs continuous |
| 6 | In-situ or manipulated | Whether the variable reported is from an in-situ observation, or from a manipulated experiment. | In-situ vs manipulated |
| 7 | Manipulation method | How the seawater chemistry is manipulated (e.g., bubbling $CO_2$ to make it more acidic, or adding solid substances to increase its alkalinity, etc.) | n.a. |
| 8 | Measured or calculated | Whether the variable is measured in-situ, or calculated from other variables. For example, salinity calculated from chlorinity is not a calculated variable, but pH calculated from DIC and TA is. | Measured vs. calculated |
| 9 | Calculation method and parameters | Information about how the variable was calculated, e.g., using a Matlab version of the CO2SYS with the dissociation constants of Lueker et al., 2000 for carbonic acid, etc. | n.a. |

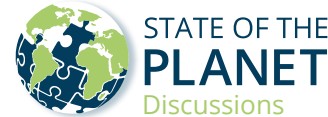

| 10 | Sampling instrument | Instrument that is used to collect water samples, or deploy sensors, etc. For example, a Niskin bottle, pump, CTD, etc is a sampling instrument. See Table 8 for a list of controlled vocabularies for this element. | Table 8 |
|---|---|---|---|
| 11 | Sampling method | Additional information describing how the sample was collected,  including more details of the instrument (such as the make, model, resolution, precisions, etc). | n.a. |
| 12 | Analyzing instrument | Instrument that is used to analyze the water samples collected with the 'sampling instrument', or the sensors that are mounted on the 'sampling instrument' to measure the water body continuously. For example, a coulometer, winkler titrator, spectrophotometer, pH meter, thermosalinograph, oxygen sensor, YSI Multiparameter Meter, etc is an analyzing instrument. See Table 8 for a list of controlled vocabularies for this element. | Table 8 |
| 13 | Analyzing method | Additional information describing how the sample was analyzed, including more details of the instrument (such as the make, model, resolution, precisions, etc). | n.a. |
| 14 | Calibration info | Information about how and when the sensor was calibrated. | n.a. |
| 15 | Field replication info | Repetition of sample collection and measurement, e.g., triplicate samples. | n.a. |
| 16 | QC steps taken | What QC steps have been taken to improve the quality of the data. | n.a. |
| 17 | Uncertainty | Uncertainty of the results (e.g., 1%, 2 μmol/kg), or a description of the uncertainties involved in this method. | n.a. |
| 18 | Weather quality or climate quality | The climate quality objective requires that a change in the dissolved carbonate ion concentration to be estimated at a particular site with a relative standard uncertainty of 1%. The carbonate ion concentration is calculated from two of the four carbonate system parameters and implies an uncertainty of approximately 0.003 in pH; of 2 μmol kg$^{-1}$ in measurements of TA and DIC; and a relative uncertainty of about 0.5% in the $p$CO$_2$. The weather objective requires the carbonate ion concentration (used to calculate saturation state) to have a relative standard uncertainty of 10%. This implies an uncertainty of approximately 0.02 in pH; of 10 μmol kg$^{-1}$ in measurements of TA and DIC; and a relative uncertainty of about 2.5% in $p$CO$_2$. Newton et al. (2015). | Weather vs Climate |
| 19 | QC flag scheme | Describe what the quality control flags stand for, e.g., 2 = good value, 3 = questionable value, 4 = bad value. | n.a. |





| 20 | Biological subject | For biological variables, please state the taxonomy (a taxon or a community), upon which the variable is studied. For example, if you study the growth rate of a certain species of salmon. The "variable/parameter" is growth rate, and "Biological subject" is that species of salmon. You could group/capture organismal data in three forms: taxonomic, functional, and phylogenetic. | Catalogue of Life (COL): https://www.catalogueoflife.org/, Integrated Taxonomic Information System (or IT IS): https://www.itis.gov/index.html, World Register of Marine Species (or WoRMS): http://marinespecies.org/, or Paleobiology Database (PBDB): https://paleobiodb.org/classic/ |
|----|--------------------|----------------------------------------------------------------------------------------------------------------------------------------------------------------------------------------------------------------------------------------------------------------------------------------------------------------------------------------------------------------------|---------------------------------------------------------------------------------------------------------------------------------------------------------------------------------------------------------------------------------------------------------------------------------------------------------------------|
| 21 | Species ID | A persistent, unique code as an identifier for a taxonomic entry. For example AphiaID in WoRMS, or the Taxonomic Serial Number (TSN) in ITIS. | See above |
| 22 | Life stage | Organisms often go through several distinct stages during their development. This can be any stages like egg, embryo, larva, juvenile, and adult. | SeaDataNet development stage: https://vocab.nerc.ac.uk/collection/S11/current/ |
| 23 | Other detailed information | Other detailed information about how the variable was collected and measured. | n.a. |
| 24 | Method reference | Citation for the method. It is recommended to use https://www.citationmachine.net to generate the citation. | n.a. |
| 25 | Researcher who measured this variable | The name and affiliation of the investigator responsible for measuring this variable. | ORCID: https://orcid.org/. |

## 4 Controlled vocabularies

For OAE data management, metadata elements that should be supported with controlled vocabularies include observed properties (e.g., DIC, TA, dissolved oxygen), observation or study types (e.g., surface underway, time-series), platforms (e.g., research vessels), sea names, instruments, people, institutions, countries, etc. For platforms, refer to the International

Council for the Exploration of the Sea (ICES): https://vocab.ices.dk/?ref=315. For sea names, it is recommended to use the SeaDataNet C16 list (sea areas): https://vocab.seadatanet.org/v_bodc_vocab_v2/search.asp?lib=C16. For countries, use the SeaDataNet C32 list (International Standards Organisation Countries): https://vocab.seadatanet.org/v_bodc_vocab_v2/search.asp?lib=C32. For investigator names, it is recommended to use the list as managed by ORCID: https://orcid.org/. For institutions, refer to the Research Organization Registry (ROR):

https://ror.org/. Another two groups of controlled vocabularies related to OAE studies are presented here: (a) types of OAE studies (Table 4), and (b) types of alkalinization (Table 5).

**Table 4.** Controlled vocabularies for major types of OAE studies. NVS is short for NERC Vocabulary Server (NVS) (link: https://vocab.nerc.ac.uk/). SDN is short for SeaDataNet. "n.a." is short for "not available". Refer to Table 1 for more





information about some of these study types. For the latest version of this list, refer to:

https://www.ncei.noaa.gov/access/ocean-carbon-acidification-data-system/vocabularies/observation-types.html.

| No. | OAE study types | NVS term [ID] and link |
|-----|-----------------|------------------------|
| 1 | Profile | Water bottle stations [SDN:C77::H09] (http://vocab.nerc.ac.uk/collection/C77/current/H09/); CTD stations [SDN:C77::H10] (http://vocab.nerc.ac.uk/collection/C77/current/H10) |
| 2 | Surface underway | Surface measurements underway (T,S) [SDN:C77::H71] (http://vocab.nerc.ac.uk/collection/C77/current/H71/) |
| 3 | Time-series | n.a. |
| 4 | Laboratory experiments | n.a. |
| 5 | Pelagic mesocosms | n.a. |
| 6 | Benthic mesocosms | n.a. |
| 7 | Field experiments | n.a. |
| 8 | Natural analogues | n.a. |
| 9 | Model output | n.a. |

**Table 5.** Controlled vocabularies for major types of alkalinization (based on Renforth and Henderson, 2017; Caserini et al., 2022). For the latest version of this list, refer to: https://www.ncei.noaa.gov/access/ocean-carbon-acidification-data-system/vocabularies/alkalinization-types.html.

| No. | Types of alkalinization | Chemical formula | OAE mechanism |
|-----|-------------------------|------------------|---------------|
| 1 | Anorthite | $CaAl_2Si_2O_8$ | $CaAl_2Si_2O_8 + 2CO_2 + 3H_2O \rightarrow Ca^{2+} + 2HCO_3^- + Al_2Si_2O_5(OH)_4$ |
| 2 | Brucite | $Mg(OH)_2$ | $Mg(OH)_2 + 2CO_2 + H_2O \rightarrow Mg^{2+} + 2HCO_3^-$ |
| 3 | Calcite | $CaCO_3$ | $CaCO_3 + CO_2 + H_2O \rightarrow Ca^{2+} + 2HCO_3^-$ |
| 4 | Calcium oxide | $CaO$ | $CaO + H_2O \rightarrow Ca(OH)_2$; $Ca(OH)_2 + 2CO_2 \rightarrow Ca^{2+} + 2HCO_3^-$ |
| 5 | Dolomite | $CaMg(CO_3)_2$ | $CaMg(CO_3)_2 + 2CO_2 + 2H_2O \rightarrow Ca^{2+} + Mg^{2+} + 4HCO_3^-$ |
| 6 | Forsterite | $Mg_2SiO_4$ | $Mg_2SiO_4 + 4CO_2 + 4H_2O \rightarrow 2Mg^{2+} + 4HCO_3^- + H_4SiO_4$ |
| 7 | Magnesite | $MgCO_3$ | $MgCO_3 + CO_2 + H_2O \rightarrow Mg^{2+} + 2HCO_3^-$ |
| 8 | Olivine | $(Mg, Fe)_2SiO_4$ | $Mg_2SiO_4 + 4CO_2 + 4H_2O \rightarrow 2Mg^{2+} + 4HCO_3^- + H_4SiO_4$ <br> $Fe_2SiO_4 + 4CO_2 + 4H_2O \rightarrow 2Fe^{2+} + 4HCO_3^- + H_4SiO_4$ |
| 9 | Periclase | $MgO$ | $MgO + 2CO_2 + H_2O \rightarrow Mg^{2+} + 2HCO_3^-$ |



| 10 | Portlandite | Ca(OH)$_2$ | Ca(OH)$_2$ + 2CO$_2$ + H$_2$O → Ca$^{2+}$ + 2HCO$_3^-$ |
| 11 | Sodium hydroxide | NaOH - brine | NaOH + CO$_2$ → Na$^+$ + HCO$_3^-$ (electrochemical weathering) |
| 12 | Sodium carbonate | Na$_2$CO$_3$ | Na$_2$CO$_3$ + CO$_2$ → 2Na$^+$ + 2HCO$_3^-$ |
| 13 | Sodium bicarbonate | NaHCO$_3$ | NaHCO$_3$ → Na$^+$ + HCO$_3^-$ |


Controlled vocabularies play a crucial role in data management, enabling researchers to describe their data in a standardized and precise way. Among the various types of controlled vocabularies, observed properties are particularly important, as they describe the measurable characteristics of a survey or experiment. However, observed properties also pose some challenges, as the terms used to describe them can be highly specialized and context-dependent. For example, different prefixes and

postfixes may be added to the same basic term, resulting in a proliferation of narrow and highly specific terms (see examples in Table 6). This can make it difficult to find the right term for a given purpose, and can also lead to inconsistencies and confusion. Furthermore, different communities may use slightly different terms to describe the same property, or may have different conventions for expressing units and dimensions.

**Table 6.** Variables related to total dissolved inorganic carbon contents (DIC) within the Climate and Forecast (CF)
conventions (https://cfconventions.org/).

| Standard Name | Canonical Units |
|---|---|
| mole_concentration_of_dissolved_inorganic_carbon_abiotic_analogue_in_sea_water | mol m$^{-3}$ |
| mole_concentration_of_dissolved_inorganic_carbon_in_sea_water | mol m$^{-3}$ |
| mole_concentration_of_dissolved_inorganic_carbon_natural_analogue_in_sea_water | mol m$^{-3}$ |
| moles_of_dissolved_inorganic_carbon_per_unit_mass_in_sea_water | mol kg$^{-1}$ |
| ocean_mass_content_of_dissolved_inorganic_carbon | kg m$^{-2}$ |
| tendency_of_mole_concentration_of_dissolved_inorganic_carbon_in_sea_water_due_to_biological_processes | mol m$^{-3}$ s$^{-1}$ |
| tendency_of_ocean_mole_content_of_dissolved_inorganic_carbon | mol m$^{-2}$ s$^{-1}$ |
| tendency_of_ocean_mole_content_of_dissolved_inorganic_carbon_due_to_biological_processes | mol m$^{-2}$ s$^{-1}$ |

The current setup makes it necessary to create multiple variations of the same property, defeating the purpose of controlled vocabularies. Moving forward, it is important to develop clear guidelines and standards to foster collaboration and communication among different communities. Specifically, it is recommended to manage controlled vocabularies for

different types of information separately. Imagine the CF convention only has one clean term called "Dissolved inorganic carbon", with a preferred unit of "µmol/kg". The list will be significantly shorter, and each of the terms will be much broadly used. It would also be much more cost-effective to manage a shorter list. Ideally, such vocabulary development efforts should be driven by the scientific community to ensure their accuracy and the developed list will conform to the needs and




preferences of their research. Before those clean lists were developed, it is recommended to use the list as documented in
Table 1 of Jiang et al. (2015) for the purpose of standardizing observed properties.

Additionally, two new types of controlled vocabularies were introduced. In the metadata template described by Jiang et al. (2015), a metadata section called platform is used to document the platform information. This section contains information such as platform name, ID, type, owner, and country. Of these elements, the platform type could play an important role when it comes to data search purposes. SeaDataNet manages a similar list called "seavox platform categories" (L06) for this
purpose. However, it does not cover all the terms the OAE research needs. In this Chapter, we introduced a new list for this purpose (Table 7). Similarly, SeaDataNet has a list called "device categories" (L05) for the types of instruments, although it does not have all the needed terms for OAE research. Table 8 lists instruments that are most likely used in this field.

**Table 7.** Controlled vocabularies for platform types. NVS is short for NERC Vocabulary Server (NVS) (link: https://vocab.nerc.ac.uk/). SDN is short for SeaDataNet. For the latest version of this list, refer to:
https://www.ncei.noaa.gov/access/ocean-carbon-acidification-data-system/vocabularies/platform-types.html.

| No. | Platform type | NVS term [ID] and link | Description |
|---|---|---|---|
| 1 | Research vessel | Research vessel [SDN:L06::31] http://vocab.nerc.ac.uk/collection/L06/current/31/ | A research vessel is a specialized type of ship or boat that is designed and equipped for oceanographic research. It often has autonomous sensors onboard and laboratories with scientific equipment for analyzing samples, and various other facilities to support research operations at sea. |
| 2 | Ship of opportunity (SOOP) | Vessel of opportunity [SDN:L06::32] http://vocab.nerc.ac.uk/collection/L06/current/32/ | Ships of opportunity (SOOP) are not specifically designed for oceanographic research but are used to collect scientific data from autonomous sensors opportunistically. They can be cargo vessels, container ships, or other types of vessels that travel predetermined routes across the ocean. |
| 3 | Mooring | Mooring [SDN:L06::48] http://vocab.nerc.ac.uk/collection/L06/current/48/ | A mooring is a collection of instruments used to measure oceanographic variables over an extended period of time at a fixed station. These mooring systems typically comprise a surface or subsurface buoy, to which the instruments are affixed, and a weighted anchor connected by a line. |
| 4 | Drifting buoy | drifting surface float [SDN:L06::42] http://vocab.nerc.ac.uk/collection/L06/current/42/ | Drifting buoys are devices that float on the ocean surface, allowing them to follow the current. Typically, these buoys are equipped with a "drogue" — a device like a parachute or sheet — which enables them to be dragged along by the current. |
| 5 | Argo float | Drifting subsurface profiling float [SDN:L06::46] http://vocab.nerc.ac.uk/collection/L06/current/46/ | Argo floats are a type of profiling float, consists of a cylindrical body that contains sensors for measuring ocean properties and inflatable bladders that allow the float to change its buoyancy and move up and down through the water column. Argo floats drift with ocean currents and surface periodically to transmit data via satellite. |



| 6 | Glider | Glider [SDN:L06::6A] http://vocab.nerc.ac.uk/collection/L06/current/6A/ | Ocean gliders are a type of autonomous underwater vehicle (AUV) that moves through the water using changes in buoyancy and wings to control its movement. Unlike traditional AUVs, which use propellers to move through the water, ocean gliders move by changing their buoyancy and using wings to "glide" through the water. |
| 7 | Saildrone | Autonomous surface water vehicle [SDN:L06::3B] http://vocab.nerc.ac.uk/collection/L06/current/3B/ | Saildrones are a type of autonomous surface vessels (ASVs) that can travel long distances over extended periods of time. These environmentally friendly ocean drones are powered exclusively by the wind (for propulsion) and solar (for the onboard instruments). |

**Table 8.** Controlled vocabularies for instrument types. NVS is short for NERC Vocabulary Server (NVS) (link: https://vocab.nerc.ac.uk/). SDN is short for SeaDataNet. "n.a." is short for "not available". For the latest version of this list, refer to: https://www.ncei.noaa.gov/access/ocean-carbon-acidification-data-system/vocabularies/instrument-types.html.

| No. | Instrument type | NVS term [ID] and link | Description |
|-----|-----------------|------------------------|-------------|
| 1 | CTD Rosette | n.a. | A CTD rosette consists of a metal frame that houses a collection of sensors and water sampling bottles (e.g., Niskin). |
| 2 | CTD sensor | CTD [SDN:L05::130] http://vocab.nerc.ac.uk/collection/L05/current/130/ | The acronym CTD stands for Conductivity, Temperature, and Depth, which are the three primary variables measured by a CTD sensor. |
| 3 | Niskin bottle | Discrete water samplers [SDN:L05::30] http://vocab.nerc.ac.uk/collection/L05/current/30/ | A Niskin bottle is a type of sampling device used in oceanography to collect water samples at different depths. It is named after the inventor, Shale Niskin, who developed the device in the 1960s. |
| 4 | Flow-through system | Continuous water samplers [SDN:L05::31] http://vocab.nerc.ac.uk/collection/L05/current/31/ | A flow-through system on a research vessel or ship of opportunity is a system designed to continuously pump seawater from the ocean into the laboratory for scientific research. |
| 5 | Thermosalinograph | Thermosalinographs [SDN:L05::133] http://vocab.nerc.ac.uk/collection/L05/current/133/ | A Thermosalinograph (TSG) is an instrument used to measure seawater temperature and salinity. |
| 6 | Salinometer for discrete salinity measurement | Salinometers [SDN:L05::LAB30] http://vocab.nerc.ac.uk/collection/L05/current/LAB30/ | Salinometers work based on the principle of conductivity. They measure the electrical conductivity of the water, which is directly related to its salinity. |





| 7 | DIC analyzers based on Coulometers | n.a. | DIC coulometers are widely used in oceanographic research to measure the concentration of dissolved inorganic carbon in seawater samples. They are often coupled with computer-controlled automated dynamic headspace analyzers that extracts total carbon dioxide from seawater using Single-Operator Multiparameter Metabolic Analyzers (SOMMAs) |
|---|---|---|---|
| 8 | DIC analyzers based on $CO_2$ gas detectors | n.a. | DIC analyzers based on a $CO_2$ gas detector including Non-dispersive infrared absorption (NDIR) (e.g., Licor LI-850), Cavity Enhanced Absorption Spectroscopy (e.g., Licor's LI-7815), and Cavity Ring-Down Spectroscopy (CRDS) (e.g., Picarro G2131i) detectors. |
| 9 | Autonomous DIC sensor | Inorganic carbon analysers [SDN:L05::86] http://vocab.nerc.ac.uk/collection/L05/current/86/ | Autonomous dissolved inorganic carbon (DIC) sensors are devices that can measure the concentration of DIC in seawater or other natural waters in situ, without the need for manual sampling and laboratory analysis. |
| 10 | Alkalinity titrator | Titrators [SDN:L05::LAB12] http://vocab.nerc.ac.uk/collection/L05/current/LAB12/ | An alkalinity titrator is a device used to measure the total alkalinity of a seawater by titration. |
| 11 | Autonomous TA sensor | n.a. | Autonomous total alkalinity (TA) sensors are devices that can measure the concentration of TA in seawater or other natural waters in situ, without the need for manual sampling and laboratory analysis. |
| 12 | Showerhead equilibrator | Equilibrators [SDN:L05::EQUIL] http://vocab.nerc.ac.uk/collection/L05/current/EQUIL/ | This type of equilibrator works by spraying seawater into a gas chamber, allowing the $CO_2$ in the water to equilibrate with a gas mixture in the chamber. |
| 13 | Floating air-water equilibrator | Equilibrators [SDN:L05::EQUIL] http://vocab.nerc.ac.uk/collection/L05/current/EQUIL/ | An "h"-shaped bubble equilibrator assembly commonly used in MAPCO2 systems on moorings. For more information, refer to Friederich et al. (1995). |
| 14 | Membrane equilibrator | Equilibrators [SDN:L05::EQUIL] http://vocab.nerc.ac.uk/collection/L05/current/EQUIL/ | While seawater is passed through a membrane, $CO_2$ in the water diffuses across the membrane and equilibrates with the gas mixture, which is then analyzed to determine the $CO_2$ concentration. |
| 15 | Flask for discrete carbon dioxide measurement | n.a. | Such flasks are typically made of glass and have a capacity of around one liter. Seawater samples are collected from a specific depth using a Niskin bottle or other sampling device and transferred to the flask without exposing them to the air. The flask is then sealed with a stopper and transported to the laboratory for analysis. |

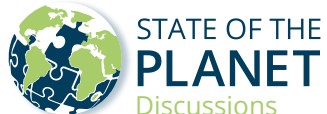

| 16 | Spectrophotometer | Spectrophotometers [SDN:L05::LAB20] http://vocab.nerc.ac.uk/collection/L05/current/LAB20/ | A spectrophotometer is a scientific instrument used to measure the amount of light absorbed or transmitted by a sample. It is commonly used for high quality pH measurements. |
|----|-------------------|----------------------------------|----------------------------------|
| 17 | Handheld pH spectrophotometer | n.a. | One example of a handheld pH spectrophotometer is the "pHyter". Refer to Pardis et al. (2022) for more details. |
| 18 | pH electrode | pH sensors [SDN:L05::355] http://vocab.nerc.ac.uk/collection/L05/current/355/ | A pH electrode, sometimes referred to as a pH probe or pH sensor, is a device used to measure the pH of a solution. |
| 19 | Sea-Bird SeaFET | Satlantic SeaFET V1 pH sensor [SDN:L22::TOOL1292] http://vocab.nerc.ac.uk/collection/L22/current/TOOL1292/ <br><br> Satlantic SeaFET V2 {Sea-Bird Scientific} (shallow) pH sensor [SDN:L22::TOOL1293] http://vocab.nerc.ac.uk/collection/L22/current/TOOL1293/ | Sea-Bird SeaFET is a type of oceanographic instrument that is used to measure the pH of seawater in real-time. |
| 20 | Oxygen titrator | Titrators [SDN:L05::LAB12] http://vocab.nerc.ac.uk/collection/L05/current/LAB12/ | An oxygen titrator is a device used to measure the concentration of dissolved oxygen in a water sample, as required for the Winkler method. |
| 21 | Oxygen sensor | Dissolved gas sensors [SDN:L05::351] http://vocab.nerc.ac.uk/collection/L05/current/351/ | An oxygen sensor or probe or sond, is an electronic device that measures the concentration of dissolved oxygen in the ocean. |
| 22 | Sea-Bird SeapHOx | Sea-Bird SBE SeapHOx V2 conductivity, temperature, pH, and dissolved oxygen system [SDN:L22::TOOL1895] http://vocab.nerc.ac.uk/collection/L22/current/TOOL1895/ | Sea-Bird SeapHOx is a type of oceanographic instrument that measures both the pH and dissolved oxygen concentration of seawater in real-time. |
| 23 | YSI | YSI Incorporated [SDN:B75::ORG00475/] http://vocab.nerc.ac.uk/collection/B75/current/ORG00475/ | YSI (Yellow Springs Instruments) is a company that produces a variety of water quality monitoring instruments. The YSI sensors are designed to measure a wide range of parameters, including temperature, salinity, and dissolved oxygen. |
| 24 | Nutrient analyzer | Nutrient analysers [SDN:L05::181] http://vocab.nerc.ac.uk/collection/L05/current/181/ | A nutrient analyzer is a device used to measure the concentration of nutrients, such as nitrate, nitrite, ammonium, phosphate, and silicate, in water samples. |



| 25 | Fluorometers | Fluorometers [SDN:L05::113] http://vocab.nerc.ac.uk/collection/L05/current/113/ | Fluorometers can detect photosynthetic pigments such as chlorophyll by transmitting an excitation beam of light and detecting the light fluoresced by the pigments in a sample. |
|---|---|---|---|
| 26 | High performance liquid chromatography (HPLC) | High performance liquid chromatographs [SDN:L05::LAB11] http://vocab.nerc.ac.uk/collection/L05/current/LAB11/ | High performance liquid chromatography (HPLC) is a powerful analytical technique used to separate, identify, and quantify individual components in a liquid mixture. |
| 27 | Acoustic Doppler Current Profiler (ADCP) | Current profilers [SDN:L05::115] http://vocab.nerc.ac.uk/collection/L05/current/115/ | Acoustic Doppler Current Profiler (ADCP), is a type of instrument used to measure water currents in oceans, rivers, and other bodies of water. |
| 28 | Mass spectrometers | Mass spectrometers [SDN:L05::LAB16] http://vocab.nerc.ac.uk/collection/L05/current/LAB16/ | A mass spectrometer is an analytical instrument used to measure and identify the mass and abundance of atoms and molecules in a sample. |
| 29 | Isotope ratio mass spectrometers (IRMS) | Isotope ratio mass spectrometers [SDN:L05::LAB48] http://vocab.nerc.ac.uk/collection/L05/current/LAB48/ | An isotope ratio mass spectrometer (IRMS) is a scientific instrument used to measure the isotopic composition of a sample. |
| 30 | Barometric pressure sensor | Meteorological packages [SDN:L05::102] http://vocab.nerc.ac.uk/collection/L05/current/102/ | A barometric pressure sensor is a device that measures atmospheric pressure, which is the pressure exerted by the weight of the Earth's atmosphere. |
| 31 | Microscopes | Optical microscopes [SDN:L05::LAB05] http://vocab.nerc.ac.uk/collection/L05/current/LAB05/ | A microscope is an instrument used to observe and magnify objects that are too small to be seen by the naked eye. |
| 32 | Scanning Electron Microscopes | Electron microscopes [SDN:L05::LAB07] http://vocab.nerc.ac.uk/collection/L05/current/LAB07/ | A scanning electron microscope (SEM) is a type of microscope that uses a focused beam of electrons to create high-resolution images of the surface of a specimen. |
| 33 | Biological trawl | Pelagic trawl nets [SDN:L05::23] http://vocab.nerc.ac.uk/collection/L05/current/23/ | A biological trawl is a type of fishing net that is towed behind a boat to collect marine organisms from the water column. |
| 34 | Phytoplankton net | Plankton nets [SDN:L05::22] http://vocab.nerc.ac.uk/collection/L05/current/22/ | Phytoplankton net is used to collect phytoplankton, which are microscopic unicellular autotrophic organisms that form the base of the marine food web. |





| 35 | Zooplankton net | Plankton nets [SDN:L05::22] http://vocab.nerc.ac.uk/collection/L05/current/22/ | Zooplankton net is used to collect zooplankton, which are microscopic animals that feed on phytoplankton and are important prey for many marine organisms. |
|----|----|----|----|
| 36 | Flow cytometers | Flow cytometers [SDN:L05::LAB37] http://vocab.nerc.ac.uk/collection/L05/current/LAB37/ | A flow cytometer is a scientific instrument used to sort and count cells or particles in a liquid suspension based on their fluorescence and other physical properties. |
| 37 | eDNA sampler | n.a. | Environmental DNA (eDNA) samplers: used to collect and analyze genetic material shed by marine organisms, which can provide information about their distribution, abundance, and diversity. |

## 5 Data citation

For oceanographic research, data citation commonly includes information such as a list of ordered authors, publication year, title, version, repository, and persistent identifier (e.g. DOI or URL) for the data set. Here is an example of a good data citation "*Feely, Richard A.; Alin, Simone R.; Hales, Burke; Johnson, Gregory C.; Juranek, Laurie W.; Byrne, Robert H.; Peterson, William T.; Goni, Miguel; Liu, Xuewu; Greeley, Dana (2015). Dissolved inorganic carbon, total alkalinity, pH, temperature, salinity and other variables collected from profile and discrete sample observations using CTD, Niskin bottle, and other*

*instruments from R/V Wecoma in the U.S. West Coast California Current System during the 2011 West Coast Ocean Acidification Cruise (WCOA2011) from 2011-08-12 to 2011-08-30 (NCEI Accession 0123467). Version 3.3. NOAA National Centers for Environmental Information. Dataset. https://doi.org/10.7289/v5jq0xz1. Accessed on 2023-03-15.*"

  There are three important considerations when it comes to minting DOIs for datasets. Firstly, it is advisable to avoid using different DOIs for different versions of the same dataset. Instead, it is recommended to mint one DOI that covers all versions

of the dataset. This approach ensures that users with a DOI can always access the latest version of the dataset, as well as any historical versions. To differentiate between versions, the citation for the dataset should include its version information. Secondly, it is crucial to wait until the dataset is published in a long-term archive with a stable link before minting a DOI. A DOI is only as reliable as the link it resolves to, so it is essential to ensure that the link is stable and will not change in the future. If the link changes later on, the DOI will become broken. Thirdly, it is important to ensure that only one DOI is assigned

to a dataset in the data flow. It is not uncommon for a dataset to be submitted to a data assembly center (DAC), and be forwarded to another DAC for different purposes later on. To avoid the risk of confusing users with multiple versions of the same dataset in different places, it is essential to make sure that only one DOI is minted for the authoritative version of the dataset. According to the NOAA plan to increase Public Access to Research Results (PARR) (NOAA, 2015), only NOAA National Data Centers are authorized to mint DOIs for NOAA funded datasets.



## 6 Data repositories

Instead of attempting to create new data repositories specifically for OAE data, it is recommended to upgrade existing OA data repositories to accommodate OAE data. After all, most types of OAE data are very similar to OA data. Ideally, scientists should only need to submit their data once, and all distributed DACs act as regional nodes, thereby contributing to the availability of ocean carbon and acidification data through a centralized data portal. Achieving this goal requires the provision of standardized metadata to the search engine of the agreed-upon one-stop portal. The most recent data management initiative by the U.N. Ocean Acidification Research for Sustainability (OARS) recommends the use of the GOA-ON Portal as the envisioned one-stop OA data portal. Once implemented, users can use the GOA-ON Portal to search for and access all ocean carbon and acidification data of a specific type. Upon discovering a dataset through the Portal, the user can then return to the respective regional DAC to access the data files and locate pertinent metadata information. In order for the above mentioned federated system to work, each of the DACs would need to meet these standards as below:

1. A long-term archive to ensure uninterrupted data access into the future.

2. Strict version control capabilities, preserving all historical versions of a dataset on a permanent basis.

3. An online submission interface, enabling users to prepare metadata in a machine-readable format, and to upload data files. Ideally, it should incorporate a user profile management interface, enabling users to keep track of all historical submissions, and resume a submission at a later time.

4. A community-driven common metadata template to support the management of comprehensive metadata information needed for ocean alkalinity enhancement research.

5. Metadata is stored in:

   a. A user-friendly interface for metadata readability (e.g., HTML).

   b. A machine-readable format to facilitate machine-to-machine interoperability (e.g., XML, SQL).

5. Controlled vocabularies are utilized to various aspects of the metadata to ensure  easy machine-to-machine metadata exchange, and successful data findability.

6. Data citation with permanent digital object identifiers (DOIs).

7. An existing mechanism to share standardized metadata with the search engine of the agreed upon data portal.

Before such a system is established, it is recommended to share a copy of the data with the Ocean Carbon and Acidification Data System (OCADS) at NOAA's National Centers for Environmental Information (NCEI), or other qualified DACs to ensure timely inclusion into data products, e.g., the Surface Ocean $CO_2$ Atlas (SOCAT), and Global Ocean Data Analysis Product Version 2 (GLODAPv2). OCADS manages a wide range of ocean carbon and acidification data, including chemical, physical, and biological observations collected from research vessels, ships of opportunity, and uncrewed platforms, as well as laboratory experiment results, and model outputs (Jiang et al., 2023). It has an established setup to channel incoming datasets to existing data products, such as SOCAT and GLODAPv2. OCADS welcomes submissions from scientists and institutions around the world. Follow this link to access the homepage of OCADS: https://www.ncei.noaa.gov/products/ocean-carbon-



acidification-data-system. Genetics or eDNA data is an exception and should be sent to the National Center for Biotechnology Information (NCBI).

In Europe, in situ OA data are typically submitted to National Oceanographic Data Centres (NODCs) along with other types of measurements. Some research groups may also submit their OA data to specialized data assembly centers like SOCAT or publish their experimental data through data publishers like Pangaea. Government monitoring agencies in Northern Europe typically send their OA data to the International Council for the Exploration of the Sea (ICES). Data centers may then integrate these data with other measurements in their databases using controlled vocabularies and standardized metadata elements. Since

the late 1990s, data centers and associated organizations involved in marine data collection, management and curation in European countries have collaborated as part of SeaDataNet, SeaDataNet 2, and SeaDataCloud. These projects have developed and adopted common standards for vocabularies, metadata schemas, data formats, and quality control procedures, enabling harmonization and interoperability of diverse marine data across Europe. The SeaDataNet infrastructure and common standards are critical to the operation and strengthening of key data workflows that feed into the European Marine Observation

and Data Network (EMODnet), created to support the EU's integrated maritime policies. EMODnet Chemistry generates data products and provides centralized access to data relevant to the implementation of European Union maritime policies, with OA data being one of the four main focuses alongside eutrophication, contaminants, and marine litter. However, the workflow for OA data in Europe is not yet well-established, and there is an opportunity to build a harmonized workflow from data creators to data centers to data aggregators and product creators. Collaboration between data curators, IT and semantic

specialists, and scientists can help enrich the semantic annotation of OA datasets with essential metadata information, which is needed to support OA research and monitoring efforts.

## 7 Conclusions

This chapter offers comprehensive guidelines for OAE researchers to prepare their metadata and data for submission to long-term archives. These guidelines encompass a wide range of OAE data types, including discrete bottled measurements and

autonomous measurements from surface underway and uncrewed platforms such as moorings, Saildrones, gliders, and Argo floats. Furthermore, they address physiological response studies conducted in various settings, such as laboratory experiments, mesocosms, field experiments, and natural analogues. The chapter also provides a universal metadata template and data standards tailored to each type of OAE data. Additionally, it presents controlled vocabularies for observation types, alkalinization methods, platform types, and instruments. These guidelines are also applicable to ocean acidification data.






## 8 Author contribution

L-QJ created the first draft. All authors contributed to the writing of this paper.

## 9 Competing interests

KF is collaborating with Planetary Technology, a climate-tech company, and Pro-Oceanus Systems Inc., an ocean technology company, as part of an NSERC Alliance Missions project focussed on OAE; none of the partners have made direct financial contributions to the project.

## 10 Acknowledgements

This is a contribution to the "Guide for Best Practices on Ocean Alkalinity Enhancement Research". We thank our funders the ClimateWorks Foundation and the Prince Albert II of Monaco Foundation. Thanks are also due to the Villefranche
Oceanographic Laboratory for supporting the lead authors' meeting in January 2023. We are grateful to Gwenaelle Moncoiffé of British Oceanographic Data Centre (Liverpool, United Kingdom) for her tremendous contributions to writing the sections on controlled vocabularies, and European data management activities. We thank Dr. Patrick Hogan and Dr. Scott Cross of NOAA National Centers for Environmental Information for providing excellent comments during the internal review process. Funding for L-QJ was from NOAA Ocean Acidification Program (OAP, Project ID: 21047) and NOAA National Centers for
Environmental Information (NCEI) through a NOAA Cooperative Institute for Satellite Earth System Studies (CISESS) grant (NA19NES4320002) at the Earth System Science Interdisciplinary Center (ESSIC), University of Maryland.



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
