# Peer review of "Data reporting and sharing for ocean alkalinity enhancement research"

_State of the Planet, 2023_

## Author Comment (AC2)

Jiang et al.'s *Data reporting and sharing for ocean alkalinity enhancement research* presents a framework for preparing, presenting and archiving data and metadata for ocean alkalinity enhancement (OAE) research, primarily through the use of existing controlled vocabularies and current best practices used in environmental data management. Overall, the framework presented appears comprehensive and well thought out. Below, I provide a few comments and suggestions for consideration prior to publication.

**Response:** Many thanks for reviewing this paper and for the kind words.

Minor comments:

Throughout the piece, the authors refer to their work as a "chapter". Is this the correct terminology? The Copernicus platform seems to refer to the submissions and publications as "reports", but also refers to them as "papers".

**Response:** The confusion stems from a change in editorial instructions. We have replaced chapter with paper on all occasions.

Line 84: "... it is recommended" – this is recommended by whom? A citation here would be appropriate.

**Response:** We have added two citations here: Tanhua et al. (2019) and Brett et al. (2020). See Line 87.

Line 87: fCO2 here refers to CO2 fugacity, but it may be worth mentioning that other communities use fCO2 to refer to the flux of CO2 across an interface (e.g., sea-air interface, biosphere-atmosphere interface)

**Response:** Two new sentences have been added as recommended. See Lines 91-93.

"Note that other communities may use $FCO_2$ to refer to the flux of $CO_2$ across an interface (e.g., sea-air interface, biosphere-atmosphere interface). It is recommended to use an italicized $f$ for fugacity and a capital F for fluxes."

Line 109: "... as with the other data standards". Do all data standards permit the addition and removal of relevant/irrelevant columns? A few example data standards here would be useful.

**Response:** Yes, all data standards are designed for best practice recommendation purposes only. We have removed this sentence to avoid confusion.

Box 1: "Refer to Chapter 4.6 for more context." It is not clear what this refers to. A citation and reference to the material in question would be useful.

**Response:** It has been replaced with "Fennel et al. (2023, this Guide)". All mentions of Chapters throughout the paper have also been addressed similarly.

Line 101: "Expectations will likely emerge through a community process". Is there an example of where this has occurred? What is the timeframe for such a process? What feedback mechanisms could help to ensure community buy-in in this approach?

**Response:** To simplify the topic, this sentence has been removed.

Table 3 lines 5 and 6: Discrete vs. continuous and in-situ vs. manipulated. Are these the best binaries? One can imagine continuous sensor-based measurements of a manipulated experiment, but also in-situ measurements of a manipulated experiment. I tend to think of "in-situ" to be quasi-synonymous with continuous, sensor-based measurements.

**Response:** We believe these are two separate and independent concepts. *In situ* measurements can be both continuous and discrete. The same thing is true for manipulated experiments.

Table 3 line 14: Calibration info on when calibrated could be requested using ISO 8601 (yyyy-mm-dd) format.

**Response:** Following the Reviewer's suggestions, we have changed this box to "Information about how and when the sensor was calibrated (ISO 8601 format: yyyy-mm-dd)"

Table 3 Line 19: I would expect to see what QC=1 represents in the schema

**Response:** We have added "1 = not evaluated/quality unknown" to this box. For more details, check out https://doi.org/10.3389/fmars.2021.705638.

Table 8: Is "CTD chain" part of a controlled vocabulary?

**Response:** Considering this item offers little extra information about the sampling, we would like to leave it out. That said, these tables will be living documents. Based on the user needs, we would be happy to add it later.

Table 8: Lines 19, 22, 23: What are the criteria by which a tradename is included in the controlled vocabulary? Sea-Bird and YSI are included, but others are not.

**Response:** In instances where instrument of a particular brand is predominantly used within the oceanographic community, we have included their trademarked name for easy recognition.

Table 8 Line 37: Is there no NVS term or link for an eDNA sampler?

**Response:** Unfortunately, not. We tried our best to find their corresponding NVS terms. In a few cases, they are not available.

Line 316: "...it is recommended to upgrade existing OA data repositories to accommodate OAE data." It is not clear to me how big a lift this represents. Have the authors had conversations with OA data repository maintainers and/or managers?

**Response:** It will be dependent on the setup of the current data repository. In the case of NOAA/NCEI's ocean carbon and acidification data system, it will be a relatively small lift. All it takes are a few extra metadata elements, and some new OAE related controlled vocabularies.

---

## Author Comment (AC3)

Review Jiang et al.,

Jiang et al. provide very useful resources how to best store data and make them FAIR. I can follow the individual arguments and recommendations. I think this is a highly valuable contribution to the OAE best practice guide. Two major comments (and several minor comments). I hope these comments help the authors to improve their manuscript.

**Response:** Many thanks for reviewing this paper and for the kind words.

Major comments:
1) I was at times a bit lost in between all the tables and recommendations for "variables" meta data". I was often wondering: For what specific aspect would I have to look at a specific Tables? Is there a possibility to provide a flow chart that provides an overview of how the data generator should step through the various processes? Where do I start and how to I go through the process (which table is relevant for which step). Having this specific guidance would be very valuable to follow the recommendations made here.

**Response:** We agree. A new figure (Figure 1, Page 8) has been added to show a flowchart of these tables.

2) The recommendations put OAE research very close to OA research. While similarities exist between OA and OAE datasets (probably the interest in carbonate chemistry), there are also many differences (major ions, nutrients, TMs, see comment above). One could equally well say that OAE datasets is similar to iron fertilization datasets, or eutrophication datasets (which is very true for some that our group generated recently). I think the close link between OA and OAE as repeatedly emphasized here is mostly based on the historic development of the field, been driven forward recently quite a lot by the OA community. It would feel a bit like a lost opportunity to not also emphasize the clear differences that exist in the perturbation via OA and OAE, with OAE being much more multi-facetted depending on the OAE method (see Eisaman et al., this guide).

**Response:** The goal is to create a broad and integrated framework that can be universally applied to all types of oceanographic data sets, including OAE, OA, iron fertilization, eutrophication data, etc. We believe that's the best way to create a cost-effective data management system. As the Reviewer pointed out, one of the reasons we draw similarities between OA and OAE datasets are the carbonate chemistry. That said, we agree with the Reviewer in that it is our goal

to highlight any differences that exist in the perturbation via OA and OAE, as much as the differences between OAE and any other studies. That has been the focus of this paper.

A new sentence has been added as a response to the Reviewer's comment: "We note that OAE research, while historically linked to acidification research, is distinct in its application, and may require additional parameters more akin to those found in iron fertilization or other perturbation studies." See Line 207-209.

Line 18: I think there are many successful science programmes or careers without effective data management. Perhaps "effective" or "transparent"?

**Response:** Data management is important in research by enabling future users to verify the research results and make new analyses. It is especially significant in the context of oceanographic research, given the global nature of the ocean system and the necessity to consolidate data collected from various cruises. Only because some programs get away with it should not serve as an excuse for researchers to neglect data management. Unless the editor thinks otherwise, we believe that "effective" is appropriate here.

Line 22: covers or "addresses"? So far there are no such standards for OAE research, right?

**Response:** The word "covers" has been replaced with "addresses". See Line 22. That's correct based on our understanding.

Line 26: Isn't "alkalinization" one of the terms that has been seen critically? OAE instead of alkalinization?

**Response:** Here, alkalinization is used to narrowly refer to the technical step of making the ocean more alkaline, while OAE is used to cover broader topics.

Line 28: DAC is potentially a confusing abbreviation in the CDR context as it is commonly used for Direct Air Capture.

**Response:** We agree. Instances of DAC have been replaced with the full name, i.e., data assembly center, throughout this paper.

Line 48: Perhaps add a bit of context here why XLSX deserves to be highlighted. Is it because it is the excel format and many people use it? Also, isn't XLSX already there since >10 years or so? Furthermore, did it matter if I stored an excel file as XLS or XLSX? There was always open software that allowed to open any of these.

**Response:** We did not single out XLSX; instead, we mentioned various formats, including Excel, CSV, NetCDF, and more, although it's worth noting that XLSX is the primary file format we receive at NCEI for original cruise data. The main intent is to clarify that XLSX is not proprietary, as this was questioned several times during our internal review.

Line 82: also ecological response. I would change to more broadly biological, geochemical etc. Perhaps biogeochemical? I assume you refer to experimental studies more broadly but that can be much more than physiological.

**Response:** Agreed. We have added a new sentence about this. See Lines 85 and 86.

Table 1 Laboratory experiments: see major comment 2

**Response:** Table 1 is specifically designed for OAE studies. We understand the Reviewer wants to highlight the differences between OAE and OA, but we are not sure about what should be changed specifically here.

Line 112: Physiological seems incomplete (see comment above).

**Response:** We have revised this: "and more broadly biological and geochemical experimental studies". See Line 121.

Line 187 suggests that the template is applicable for mCDR, not just OAE. Was this intentional?

**Response:** "mCDR" has been replaced with "OAE". See Line 207.

Line 188: What was the rationale to have a template that can be used for OA and OAE research alike? I think those two research fields should not be married so closely together because of the many critical differences (e.g. local (OAE) vs. global (OA) perturbation, scale of the perturbation, many other perturbation types (Mg, Ca, nutrients, trace metals).

**Response:** The idea is to have a broad and integrated framework that can be applied to all types of oceanographic data. Each type of data can utilize a portion of the elements, making each of them unique. The main reason is to keep data management cost-effective.

Table 3: If I understand correctly, Table 2 is more broadly outlining the observation/experiment/model (i.e., providing metadata for the study), while Table 3 describing individual parameters, which are part of the dataset. Two questions: Aren't there some redundancies? E.g. what is the difference between

Type of study (Table 2) and Observation type (Table 3)? Second, would a quite extensive qualification as in Table 3 be required for each parameter?

**Response:** Table 2 lists sections of the overall metadata template, while Table 3 focuses on the ancillary metadata elements associated with a specific section within Table 2: the variable metadata section. A new diagram (Figure 1, Page 8) has been added to the paper to illustrate their relationship as per the Reviewer's suggestion.

"Types of study" is a lump sum of the "Observation types" appearing in all the variable metadata sections. The inclusion of a "Types of study" in Table 2 and an "Observation type" in Table 3 enables the template's versatility to cover variables collected from different observation types, at the same time allowing users to understand the whole picture of observation types for the overall data package. For instance, a data package could contain variables from underway measurements (e.g, fCO2), and discrete bottle measurements (e.g., DIC and TA), where each variable possesses its unique observation type, and the overall data package shows the types of study as underway and discrete bottle.

The answer to your Question #2 is no. The extensive list of metadata elements for each variable is made available to ensure users across different fields can document their data in full detail. Very few of the elements are mandatory.

Line 251: Alkalinization or OAE? (Same for Table 5 caption)

**Response:** We have changed it to: (b) types of source materials for OAE (Table 5). See Line 276.

Table 5: The table lists a variety of rocks or mineral types or chemicals in a rather inconsistent way. For example, Forsterite is the Mg rich end of olivine (the rock termed "dunite"), while olivine is listed extra and includes fayalite (the Fe-rich end-member). Calcium oxide is a chemical, while the material would probably be called "quicklime" here. I am not an expert on this but think Table 5 needs a major clean-up. Also, I would not call it types of alkalinization but rather something like "source materials for OAE". Perhaps refer to Eisaman et al (this guide) for consistency on this.

**Response:** We have changed the table caption accordingly. See Line 283. The contents of the table were also updated.

Table 7: I wonder to what extend mentioning of very specific tools reveals that the list is not comprehensive? Do you need to describe a bit more broadly these tools? For example, saildrone is a specific product made by a company currently used as an equivalent name for a certain tool. But isn't the point that these make

distance over the sea surface? Where would a wave-glider (with a similar purpose) fit in?

**Response:** We have added a new row for surface gliders. The use of Saildrone can help differentiate it from other uncrewed surface vehicles such as wave gliders easily. That said, we fully expect these tables to be updated later on based on the research community's preferences.

Line 317: I am not with you on this statement.

**Response:** We have removed this sentence and the one preceding it.

Line 340: I wonder what the role of GEOTRACES could be in the OAE context. Some datasets have to do less with ocean carbon than with trace metals. Most of the text is very C-centric but OAE entails more than that metals (see major comment 2).

**Response:** Any projects or data management systems are welcome to participate in the management of OAE data. Whether they are willing to participate in this effort or not is outside the scope of this paper. The purpose of this guide is to offer recommendations and guidelines that they can adopt (or not). We fully agree with the Reviewer in that OAE entails more than just ocean carbon data. For example, it can cover variables such as metals, just like any other non-carbon variables. However, the key point here is to make sure we have a set of guidelines to enable the management of these OAE variables, whether they are ocean carbon related or not. Below is the list of metadata elements for each variable. We'd be more than happen to modify the list, if the Reviewer find that trace metals entail the addition of new metadata elements.

*Variable abbreviation in data files*
*Full variable name*
*Variable unit*
*Observation type*
*Discrete or continuous*
*In-situ or manipulated*
*Manipulation method*
*Measured or calculated*
*Calculation method and parameters*
*Sampling instrument*
*Sampling method*
*Analyzing instrument*
*Analyzing method*
*Calibration info*

*Field replicate information*
*QC steps taken*
*Additional info*